# Use Dynamic Scheduling Algorithm to Assure the Quality of Educational Programs and Secure the Integrity of Reports in a Quality Management System

Yasser Ali Alshehri [1,*] and Najwa Mordhah [2]

1    Education Division of the Royal Commission at Yanbu, Computer Science and Engineering Department, Yanbu University College, Yanbu 46455, Saudi Arabia
2    Education Division of the Royal Commission at Yanbu, Management Science Department, Yanbu University College, Yanbu 46455, Saudi Arabia; mordhahn@rcyci.edu.sa
*    Correspondence: alshehriya@rcyci.edu.sa

**Abstract:** The implementation of quality processes is essential for an academic setting to meet the standards of different accreditation bodies. However, processes are complex because they involve several steps and several entities. Manual implementation (i.e., using paperwork), which many institutions use, has difficulty following up the progress and closing the cycle. It becomes more challenging when more processes are in place, especially when an academic department runs more than one program. Having *n* programs per department means that the work is replicated *n* times. Our proposal in this study is to use the concept of the Tomasulo algorithm to schedule all processes of an academic institution dynamically. Because of the similarities between computer tasks and the processes of workplaces, applying this method enhances work efficiencies and reduces efforts. Further, the method provides a mechanism to secure the integrity of the reports of these processes. In this paper, we provided an educational institution case study to understand the mechanism of this method and how it can be applied in an actual workplace. The case study included operational activities that are implemented to assure the program's quality.

**Keywords:** quality assurance; educational program; program accreditation; Tomasulo algorithm; dynamic scheduling

## 1. Introduction

Quality assurance for academic programs contains consecutive and concurrent processes [1–3]. These processes are important to meet the standards of different academic bodies (e.g., The National Center for Academic Accreditation and Evaluation NCAAA [4], The Chartered Institute of Procurement and Supply CIPS [5], The Accreditation Board for Engineering and Technology ABET [6], and The Accreditation Council for Business Schools and Programs ACBSP [7]). These processes ensure that programs take all necessary actions to meet all requirements and criteria to be recognized and accredited by these bodies. Processes help meet the quality requirements by these bodies, including the development of programs, course development, annual review of the program, and program planning.

Quality requirements for any academic program consist of too many standards and criteria. These criteria are achieved when jobs are completed and documented. These jobs should fulfill approximately one hundred criteria of the national standard NCAAA [4]. Therefore, meeting all the criteria for each program is very complicated. Automating quality procedures is even more challenging because many of these procedures are interrelated and dependent on each other. Many of these procedures go in and out of different entities. Additionally, they visit the same entity multiple times. They have to be triggered at the right time and following the correct sequence. All accreditation bodies require an on-site visit to check all documentation related to the program. Therefore, all processes must be

well preserved to be checked during the external visit. Additionally, the internal quality audit checks these documentations once per year. Furthermore, keeping the documents helps to follow up with these processes to ensure that the loop has been closed.

Working in this environment makes the work to improve the scheduling and the smooth implementation of these processes essential. This also leads to better quality control and achieving national and international accreditation with less effort. If not all of the institutions, most of the institutions use manual methods to schedule essential processes to keep their programs on the right track and maintain quality procedures. Even with some attempts to automate some institutions' services, details of the quality procedures cannot be totally included. Hence, we suggest using a dynamic scheduling technique to ensure that works are triggered, traced, and preserved. This work is inspired by the dynamic scheduling approach of the Tomasulo algorithm [8].

The Tomasulo algorithm is one method used in computers to promote dynamic scheduling and exploit instruction-level parallelism [8–10]. The concept of this method can be adapted in workplaces that involve several activities within the same process. This process goes through multiple steps that are defined in a procedure. Every step requires an action to be taken (e.g., review and approve). Therefore, we need to ensure data integrity and ensure that the process has considered the correct order from the start to the end of the process. Additionally, the process should be well preserved after completion. This allows retrieving the process at any time in the future.

In this research, we explore how we could apply the Tomasulo algorithm to control the processes of academic program operations, which can benefit in different ways. First, the Tomasulo algorithm ensures the correct sequence of the procedure to be in place. Second, it allows a department to have out-of-order execution, ensuring that the data hazards are not violated. This means that an entity should read the most updated file after being corrected or modified by all entities that interfere with the defined procedure. Additionally, the entity should add the comment or decision to the case after the contribution of all previous entities. Third, the file should be temporarily preserved during the execution of each entity. Fourth, after completing the procedure and implementing all decisions, the files should have permanent preservation. The final step helps to archive all these processes and their outcomes and can be used as evidence for the internal or external audit [11].

This paper introduces dynamic scheduling and registers renaming to the workplace by using the problem/solution method. We aim to answer the question, "how do we apply the Tomasulo algorithm to control the processes of quality procedures in educational institutes?" Specifically, we use this idea in an academic environment to ensure the program development cycle and annual operational plans. Applying this notion provides the execution of several activities in parallel without violating the data integrity. It also ensures that the data are well preserved and can be easily retrieved.

The rest of this paper is organized as follows: First, we introduce the idea of the algorithm used in this study in Section 2. We explain how the algorithm works inside the microprocessor and the similarities between the microprocessor environment operations and workplaces. Then, we explore some of the related works of the area in Section 3. Next, the methodology of this work is explained in Section 4. Next, we provide a case study of an educational institution in Section 5. Then, we provide a discussion of the case study in Section 6. Lastly, the paper is concluded in Section 7.

## 2. Background

The Tomasulo algorithm was developed by Robert Tomasulo in 1963 and was used to allow a single processor to operate sequential code instructions and exploit parallelism. The parallelism allows multiple instructions to be executed out of order. The primary aim is to achieve a low average cycle per instruction CPI (CPI < 1). The only concern with this algorithm is that some instructions may cause a stall (i.e., idle wait). The stall is caused by data dependencies, which are a significant concern when dealing with a limited number of registers (i.e., temporary storage).

In workplaces, we have different processes that can be related or unrelated. Every process has triggered activities and follows a chain of steps until the process is completed. A process cycle is defined as a period required to execute a single process (a day or a week, or a month). Activities in the workplace can be performed simultaneously, like computers. There are similarities between processes in a workplace and those in a computer. All processes in the workplace must be tracked, need temporary storage during the execution time, and must be preserved at the end of each process. Completed data need to be correctly addressed to be retrieved anytime in the future for a follow-up purpose or for auditing purposes (i.e., during the accreditation process).

In addition, many processes follow a sequential order. For example, academic programs in universities should go through a cycle of improvement. This cycle starts from receiving inputs from course reports, external assessment reviews, and student satisfaction surveys. These processes have a sequence nature that should be respected. At the same time, reports need to visit different entities (committee/department/council) and should be preserved during the review by these entities. The comments and decisions of these entities should be respected and associated with the original report. At the same time, they must be flexible in executing out of order without introducing a violation to the data integrity.

When parallelism is exploited in the computer, three types of data hazards can be introduced. First, office A and office B may both write in the same file. The Tomasulo algorithm defines this type as write after write hazard (WAW). Second, office A writes in a file that office B should be reading. This hazard is defined as write after reading (WAR). Third, office A reads from a file that office B will write in after. This type is also defined as read after write hazard (RAW). All these hazards can be avoided when the order of the process is respected.

The algorithm introduced dynamic scheduling and registered to rename techniques to avoid data hazards that may occur. In this, the processes are stacked into a queue. Then, each process is fetched in order and sent to the reservation station (the main buffer). This buffer ensures that the correct order is in place without violating any of the data hazards. Additionally, it allows out-of-order execution if there is no data dependency between them.

We believe that this algorithm can help in any workplace. Tasks in workplaces need to be issued at the right time, follow a correct procedure, and be easily checked and retrieved for follow-up or auditing purposes. Therefore, the methodology of this research applies a similar concept to the Tomasulo algorithm.

## 3. Related Works

The study [12] reviewed 202 articles from 45 journals to determine the state of research in quality management in colleges and universities. The study indicated that quality management implementation issues, quality management models, techniques and tools, and quality management dimensions are the three most common topics discussed in the sample. Overall, the study shows that the models, techniques, and quality management tools that have been successful in the industry can be relevant to higher education industries across different international areas.

Another study [13] presented a systematic breakdown of the research on the field of quality management in higher education institutes. The study defined three main levels of processes: the organizational level, the process level, and the quality management principles level. Overall, the study concluded that it is imperative and crucial to integrate the three groups to implement quality management in higher education institutions.

The results of the factor analysis in quality in higher education institutes in [14] revealed that a hierarchical model is considered most appropriate. The model entailed five primary dimensions: administrative quality, physical environment quality, core educational quality, support facilities quality, and transformative quality.

More importantly, total quality management has important superiorities in the development of education systems, and the educational institutions must meet the expectations of the public in producing qualified people who can build society and contribute to the

sustainability of the growth and development of the economy [15,16]. In this sense, based on analyses of graduates from 13 countries, the studies show that academic program aspects greatly influence the employability of the graduates. Although the quality indicators have a slight effect on the chance of finding a job, there is a significant impact on how to do the job [17].

Concerning quality control, the authors of [18] proposed to apply a balanced scorecard for quality assurance in educational management in Thailand. The results of the case study indicated a significant improvement. The study [19] developed a UML analysis for a system to assure quality in higher education. The research angle focused on the course delivery and teaching aspect and did not include managing the program in the UML analysis. The study did not consider a framework to control the quality of the institution. Additionally, the suggested model in [19] is fit to an educational institute and cannot be used for any other institution. Our study focused on a different angle to consider the quality framework of an educational program that can be applied in a different institution.

Another study [20] introduced a framework to manage quality assurance in higher educations institutions. The system aims to design a tool (containing one core module and 17 sub-modules) to avoid unnecessary and redundant tasks associated with quality in higher education institutions. In [21], the authors discuss developing algorithms that can help to achieve the standards following the requirements of ISO 9001 and ISO 21001 standards. The study was addressed at a very high level, without going into specific detail, and included a case study of a fundamental institution.

Implementing quality in higher education institutions is not an easy matter due to the complexity of the higher education environment. The study [22] investigated challenges of embedding quality into European colleges and universities. The study found that challenges are classified under three broad categories: "(1) organizational challenges that include quality system, educational system, and external stakeholders; (2) implementation challenges including execution, competency and funding challenges; (3) leadership and quality culture challenges".

Most people think that quality management is all about implementing predetermined procedures (hard quality management). However, according to [23], soft quality management (leadership, people) has a positive impact on complex quality management, and hard quality management has a direct effect on performance.

The literature indicates mixed results on the applicability of TQM principles in education, and research reveals the serious success factors of TQM in Pakistani higher education institutes. According to the results of [24], many crucial success factors make TQM complicated in higher education, such as institutional vision, leadership, type of measurement and analysis, monitoring process and evaluation, design and allocation of resources of the academic program's design and stakeholder focus.

Quality assurance might be considered a relatively new concept in quality management in Saudi higher education institutes. It may not seem to be efficiently implemented due to particular challenges [25]. According to data collected from a well-known Saudi university, the findings emphasize the importance of the clarity of the policy and procedures. Explicitly, participants show different understandings of quality assurance mechanisms, which cause inconsistency between written internal quality assurance and the actual practices. Having robust quality assurance protocols and mechanisms can be an effective remedy to motivate employees. The study [25] highlighted the importance of supportive factors to help the institution remain consistent with quality standards. Supportive factors are not limited to leadership, the awareness and orientation of employees/faculty, the institution's commitment to maintaining high standards of quality, clarity of protocols, and a system to monitor the implementation and execution of the quality procedures effectively. The study highly recommends that colleges and institutes have a mechanism to ensure clarity of the procedures and monitor quality management execution with high consistency and compliance.

On the other hand, different algorithms have been proposed for educational data (e.g., [26–29]). These studies discussed how to use data mining to predict or explain students' achievements or failures. For example, in [26], Zhang tried to find the associated factors affecting students' achievement factors, which should help educators in decision making. None of these studies discussed the educational quality system.

This study applies the Tomasulo algorithm that was designed to improve parallel execution in the 1960s [8–10]. Our interest in this study is to emulate the algorithm in a different environment. For some cases, parallel execution may be applicable. However, our concern is to ensure proper scheduling and locate information during or after the processing. So, activities are conducted, and all evidence of these activities can be retrieved. Furthermore, we should be able to follow up with activities that are not closed yet.

## 4. Methodology and Proposed Solution

In this paper, we use the problem/solution method, which efficiently answers specific questions and solves a defined problem. This study aims to provide innovative solutions to issues affecting most educational institutions around the world. As stated previously, managing quality assurance in higher educational institutions is a demand and challenging. Even though many studies have suggested models and frameworks that might fit in a particular educational institute, their suggestions cannot be generalized and applied to other institutions. This study focuses on a different angle to consider the quality control framework of an educational program by using the Tomasulo algorithm, which can be applied in other institutions.

This study aims to answer the question, "how do we apply the Tomasulo algorithm to control the processes of quality procedures in educational institutes?" In doing so, this paper proposes the Tomasulo algorithm to ensure correct scheduling and the ability to locate information during or after a quality procedure. The following discussion explains the proposed solution and how to apply it by using a case study. The case study gives a deep understanding of how the Tomasulo algorithm can be implemented to control the quality of the educational program.

Similar to processes in computers, we have four main steps for each activity that can be demonstrated through the Figure 1 and the flow chart in Figure 2. First, each process is issued from a queue of activities and from going into preparation for execution. Second, the activity is prepared in the reservation station. Some information is collected for every activity in this unit to ensure that the procedure is in place. Third, each process is executed in a concerned department and saved temporarily in registers dedicated for each operation. This stage can be conducted in a single entity, or it could involve multiple entities. Fourth, the process is preserved and saved to be retrieved in the future.

As illustrated in Figure 1, the algorithm, in the first step, fetches the first process in the queue. Second, the process goes to the reservation station. The process checks the next destination of the process and what execution is required (i.e., what to do?). Staying in the reservation station ensures that the execution of the process is in the proper order without violating the integrity of the reports. The report's integrity will be violated when the entered data do not respect the correct sequence, which is caused by a violation of the procedure used for this process. The reservation station can accommodate more of the process as long as there is more space. The process goes to the execution unit to execute a specific activity and then back to the station. It goes back and forth until the last activity of the process is executed and completed. After the final execution of the process, the outcome of the process must go to storage. We have two possible scenarios here: first, the case is finished, and the quality loop for this process is closed. Second, the case needs to be followed up, and the circle is not closed yet. In the first scenario, we need to send the file for permanent storage and use evidence for one of the standards. If the queue still has some processes to be executed, then we fetch more for the reservation station or go for termination as shown in the flow chart in Figure 2.

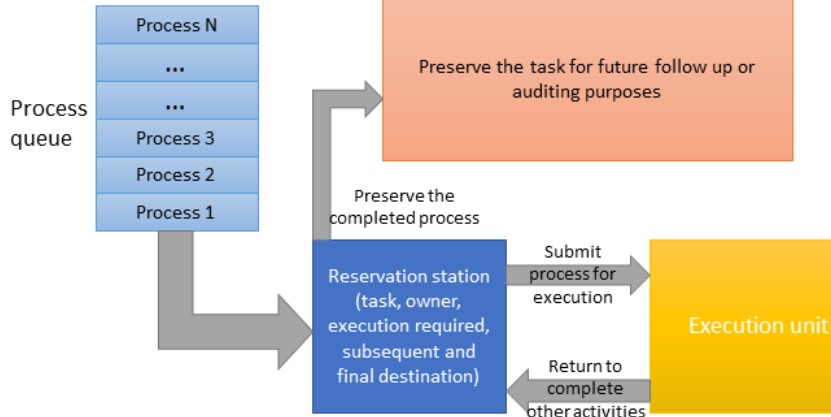

**Figure 1.** Methodology of the proposed solution.

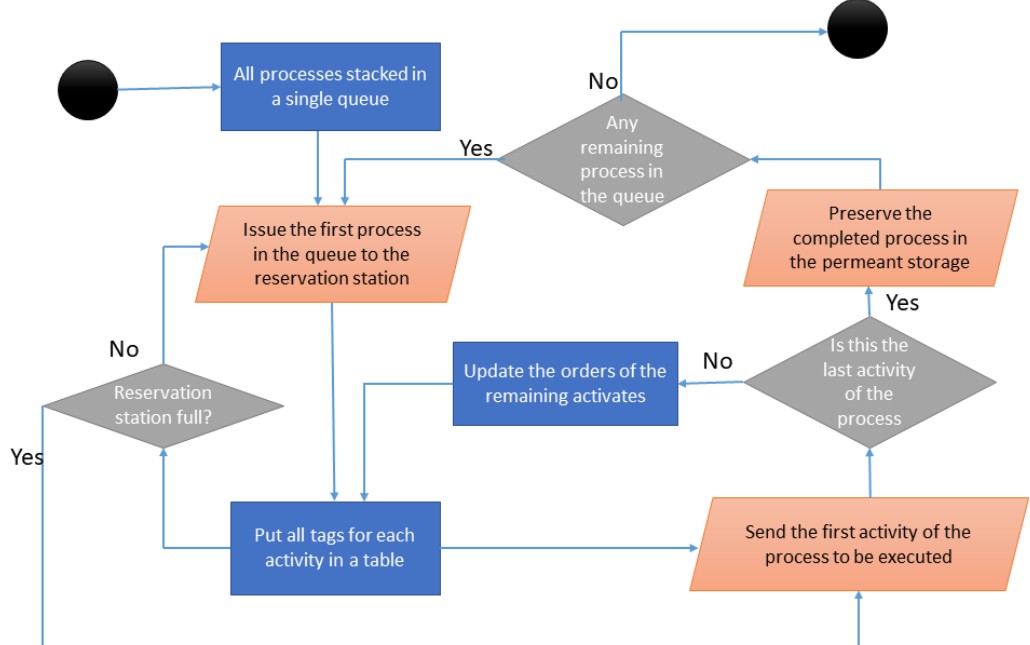

**Figure 2.** Flow chart of the proposed solution.

Before the issue stage, we need first to build the activity queue, which is similar to the instructions queue that exists in the computer as shown in Figure 1. To build the activity queue, we need to collect all operations conducted at each process. We need to know the owner of this process, the place (or the places) of execution of this process. For every operation, we need to define the office or entity that this process should visit, what type of execution is conducted, where it should go next, and whom should be responsible for the following up. All this information is gathered in a single table, which we call the reservation station table. The same process can be replicated due to the existence of multiple programs within the same department. In addition, the single process contains multiple activities to be completed (e.g., study, review, approve, notify, update, and preserve). When we need to trigger the process, the initiator needs to fetch the task from the queue, prepare it, and submit it to the reservation station.

The reservation station in the Tomasulo algorithm acts to control the flow of the program and ensure the integrity of the data (i.e., no data hazards). The Tomasulo algorithm checks first if there is a space to accommodate the instruction. Then, it fills all information for this instruction in the table. Information includes the operation and the destination of the outcome. In our approach, we had data for each activity of every process. The

reservation station should first check if there is a space to accommodate this task, the owner, and where it should go next.

As shown in Table 1, the reservation station holds the primary information to trace the process and activities within the same process. It is essential to know what type of activity should be taken and who is responsible for conducting this activity. The owner of all activities of the same process is a single entity (Entity 1). Each entry also contains the functional unit for the specific activity (i.e., the entity responsible for executing the current activity) and the next destination (i.e., the next functional unit responsible for the following activity). The last piece of information is who to follow up with for this activity.

**Table 1.** Reservation station content.

| Process ID | Activity | Operation | Owner | Functional Unit | Next Destination | Follow up with |
|---|---|---|---|---|---|---|
| Process 1 | Activity1 | Prepare | Entity1 | Entity1 | Entity2 | |
| Process 1 | Activity2 | Study | Entity1 | Entity2 | Entity3 | Entity2 |
| Process 1 | Activity3 | Approve | Entity1 | Entity3 | Entity1 | Entity3 |
| Process 1 | Activity4 | Inform | Entity1 | Entity1 | Entity4 | |
| Process 1 | Activity5 | Implement | Entity1 | Entity4 | Entity1 | Entity4 |
| Process 1 | Activity6 | Store | Entity1 | Entity1 | Preservation | Entity4 |

The execution unit is where the process is executed. It represents the offices where the process should go. The process may visit different execution units (e.g., department council, college council, the planning committee, and curriculum committee). An execution unit may handle more than one activity. These activities could be unrelated, or executing them simultaneously does not violate the sequence that both should follow. Therefore, data integrity is ensured by the reservation station, which is in charge of controlling the correct flow of all processes.

The update between the execution unit and the reservation station should go back and forth for the same process. The reservation station should update the table to omit the completed activity. When all activities of the same process are completed, the whole process should be moved for permanent storage. Outcomes of a process should be well preserved and filed for any future recall. The future recall could be due to following up and closing the loop in the quality procedure or internal or external auditing.

## 5. Case Study

In this case study, we explain how the algorithm can work in a natural work environment. We choose to describe this algorithm in an academic setting. In this environment, we deal with reviewing and developing educational programs and preparing annual operational plans. To achieve the primary operations conducted in any environment, we need three elements. First, a sound scheduling system knows when to fetch the job, execute it, and follow it until it is done. Second, we need an execution unit to receive the case at the right time and know what needs to be done. Third, we need a preservation system to archive all outcomes and be able to locate them with ease. Hence, this case study is divided into three stages.

In the first stage, we list the main processes required to manage an academic program and all activities for each process. There are many different processes that every academic department can potentially conduct. These processes are concerning academic issues (e.g., course review, course reporting, program review, and course modification), planning issues (e.g., annual business plan, budget plan, and scholarship plan), or students processes (e.g., academic advising, changing schedule, and other student services). Each program is estimated to have 35 procedures concerning planning and development, 34 procedures concerning educational issues, and 21 procedures concerning students and other issues.

However, we choose to list the most common and frequent one, conducted annually to ensure a program's quality.

The second stage focuses on fetching these processes with their activities into the reservation stations, ensuring that they are conducted correctly. The integrity of these processes was not violated, which means that every party is involved. During the execution of a process, data can be temporally preserved. Each process contains a minimum of two activities or higher. So, the process remains in the reservation station for a while, executing one activity and returning to execute another.

The third stage preserves the completed process permanently. The process proceeds after that, completing the final activity of the process as shown in Figure 2.

### 5.1. Stage 1

In this stage, all jobs are stacked in a single queue where they are to be fetched one after another (i.e., first in, first out). At the beginning of every academic year, the annual program report preparation should take place. Explicitly, each department provides a complete program report, including an assessment of curriculum, resources, faculties, student achievements, and results from various course evaluations and the support document, such as course reports, that contribute to the formulation of the program report. The program assessment and evaluation committee completes the annual program report and submits it to the Department Curriculum Committee to review it. Recommendations should be discussed and approved in the Department Council. Finally, the proposals for each program should be approved by the college council.

The Quality Department forms an Annual Program Review Team (APRT) to conduct the review according to fixed AAPR review criteria and prepare review reports. Then, all program review reports for each department are compiled and discussed by all the relevant parties.

It has to be highlighted that any change made to improve either the course or program should be evidence-based. So, feedback from stakeholders (students, faculty, employees, alumni, advisory board, etc.) should be analyzed and reported:

- Minutes of meetings of program-level committees reviewing course reports and preparing program reports.
- Minutes of meetings of program-level committees and councils approving recommended changes.
- Minutes of meetings of college/institute-level committees and councils approving recommended changes.

This cycle should consider the course reports of the previous year and student satisfaction survey outcomes. Therefore, the first activity is to complete the course report of all courses and submit it to the concerned party responsible for developing the annual program reports. Then, based on the yearly programs' recommendations, we should prepare the operational plans, including equipment, workforce, budget, and professional development needed. So, the queue of the processes should be as follows:

- **Process 1:** Action items of students' satisfaction survey activities—opening survey for all students, collect answers, analyze answers, approve, send to the concerned department to take action.
- **Process 2:** Course reports recommendations activities—submit course report, revise by the program assessment and evaluation committee, report changes needed for the course, and report course requirements, reviewed and approved by relevant authorized bodies.
- **Process 3:** Annual program report activities—obtain course reports, annual program reports, and the program assessment. The evaluation committee will write and review the findings, considering feedback from stakeholders, to suggest improvement elements on the curriculum. The program assessment and evaluation committee analyzes the resources and submits the planning specialist's need for resources.

- **Process 4:** Preparation of the operational plan activities—obtain the outcome for the resources from the annual program report, obtain the results of the key performance indicators of the last year, set new targets for the next year, confirm the budget for the next year, and confirm the professional development plan.

After building a list of all processes and activities, we need to stack them into a queue. The queue should consider all processes and the activities' timing and should be in the correct order.

*5.2. Stage 2*

The reservation station is used to control the flow of the activities and respect the order of processes. Each process occupies the reservation station as shown in Table 2. Each process's activities are reorganized inside the reservation station and queue for their orders to be executed. In this stage, the first process starts to be fetched from the line. Similar to the concept of dynamic scheduling, activities can be concurrently executed or executed out of order. Furthermore, activities can be finished out of order as long as data dependency is ensured between all activities.

**Table 2.** Reservation table.

| Process Name | Activity | Operation | Owner | Functional Unit | Next Destination | Follow up with |
|---|---|---|---|---|---|---|
| Process 1 | Activity1 | Publish survey | Quality Dept. | Quality Dept . | Quality Dept. | Quality Dept. |
| Process 1 | Activity2 | Collect and analyze | Quality Dept. | Quality Dept. | Quality Dept. | Quality Dept. |
| Process 1 | Activity3 | Submit to the Academic Dept. | Quality Dept . | Academic Dept. | Quality Dept. | Academic Dept. |
| Process 1 | Activity4 | Actions for improvement | Academic Dept. | Academic Dept. | Department Council | Academic Dept. |
| Process 1 | Activity5 | Approve actions | Academic Dept. | Academic Dept. | Academic Dept. | Academic Dept. |
| Process 2 | Activity1 | Collect the course data | Academic Dept. | Course leader | Course leader | Course leader |
| Process 2 | Activity2 | Review/Analyze | Academic Dept. | Course leader | Course leader | Course leader |
| Process 2 | Activity3 | Write report | Academic Dept. | Course leader | PEAC | PEAC |
| Process 2 | Activity4 | Approve report | Academic Dept. | Program chair | Preservation | Program chair |
| Process 3 | Activity1 | Aggregate program data | Academic Dept. | PEAC | PEAC | Program chair |
| Process 3 | Activity2 | Review/Analyze | Academic Dept. | PEAC and Curriculum Committee | PEAC | Program chair |
| Process 3 | Activity3 | Report and actions | Academic Dept. | Program committee | Academic Dept. | Program chair |
| Process 3 | Activity4 | Approve actions | Academic Dept. | Department Council | College council | Program chair |
| Process 4 | Activity1 | Prepare the department plan | Academic Dept. | Academic Dept. | Department council | Academic Dept. |
| Process 4 | Activity2 | Approve the department plan | Academic Dept. | Department council | Planning Dept. | Academic Dept. |
| Process 4 | Activity3 | Review and compile all plans | Planning Dept. | Planning Dept. | College council | Planning Dept. |
| Process 4 | Activity3 | Approve plan | Planning Dept. | College council | Preservation | Planning Dept. |

For Process 1, there are four main activities, and the owner of these activities is the quality department. An independent party should handle surveys. However, the actions and improvements should be handled by the concerned department. For Process 2, the activities of the course delivery are all completed by the academic department. Therefore, the owner and the executor of these activities are within the academic department (i.e., course leader and program chair). In Process 3, the activities rely on the outcomes of Process 1 and Process 3. Therefore, the executor and the owner of this process are the academic departments. Finally, Process 4 builds the plan for the department. After the department's plan approval, it goes to a higher level (college level), where the planning department takes the ownership and compiles it with all other college entities and raises it to the college council for final approval.

*5.3. Stage 3*

In this stage, the outcomes of all processes should be preserved. This stage is reached when all activities of a single process are executed. Then, the outcome of this particular process is sent to be archived. Finally, the file is reopened for some potential scenarios. First, some of the processes need to be used as a reference or benchmark for the following process. For example, the annual program review of this year is a benchmark for the next year.

Similarly, the key performance indicators reported this year are used to set up the target for next year. Therefore, these documents need to be accessible for review and analysis to road map the following year. Second, these documents are evidence for the compliance of the program management to the national and international. The program undergoes an internal review annually and an external review every five years. Therefore, preserving this evidence is essential for any auditing, whether for educational or non-educational institutions.

## 6. Discussion

The case study demonstrated the applicability of the novel idea of this research, which applies dynamic scheduling to the academic setting. The belief is that the flow of the processes will be enhanced, and the quality will be in control. After applying this methodology, we need to consider department chairs and program coordinators' opinions to see what this idea has provided, compared to the old style. We believe that implementing this methodology is beneficial due to the similar nature between workplaces and functions in computers.

The benefit of this methodology will be much better if it is computerized and automated. Most of the systems designed to automate the process do not fully integrate all quality procedures and requirements because developers are not set together with the program chair to understand all the quality requirements and aspects to be included in the system. In this paper, we recommend using the model of the reservation station and the Tomasulo algorithm as a baseline to construct and design the system.

As explained in the case study, each program uses almost one hundred procedures. Some of these procedures are mandatory to be executed every semester or every year. Other procedures are only used when needed. Working with this number of procedures creates numerous works, which makes scheduling vital to organize and increase work productivity.

Quality in education is supposed to develop the educational program and improve people's practice to achieve the program's goals. However, the education staff is always irritated about the volume of work is required. The amount of work sometimes is redundant due to the lack of a proper documentation process. Quality control cops are keen to see that the work complies with policy and procedures. More importantly, we need always to think about easing the process and making people willing to practice it.

The idea of this work is very generic, which means that it can be generalized to fit many institutions across the world. We need to tie this to specific case studies to explore findings in different institutions for further investigation.

## 7. Conclusions

This study introduced a novel method to help the assurance of the quality of an educational program. The methodology we adopted is derived from the well-known computer algorithm (Tomsula algorithm). This algorithm was invented in the 1960s and has been used and evolving ever since. In this research, we discussed the implementation of this methodology in an educational institution. We discussed the three stages of a process from start to finish based on the given methodology. We discussed the similarities of processes inside the computer system and any organization, which require good scheduling, good management while running and executing, and preservation. Non-educational entities can also benefit from the methodology, but only if they have a defined quality system.

This study gave an example of an academic environment and demonstrated how this methodology could be helpful. Future work should consider applying this methodology in a natural setting and report all outcomes. We need to analyze the differences between using this methodology versus not using it. Additionally, we need to think about applying the same method in a different work setting. If we use it in another setting, we need to analyze the job requirements first for this environment and then decide the order of each process, as in stage 2. Because we have different activities within each process and interrelated processes, we need to maintain healthy coordination.

**Author Contributions:** Validation, N.M.; writing—review & editing, Y.A.A. All authors have read and agreed to the published version of the manuscript.

**Funding:** This research received no external funding.

**Institutional Review Board Statement:** Not applicable.

**Informed Consent Statement:** Not applicable.

**Data Availability Statement:** Data sharing not applicable.

**Acknowledgments:** Our gratitude is due to those who contributed to establishing the quality management system in the Education Sector of the Royal Commission at Yanbu. Special thanks go to Maher Alghanim, the director-general, Mahdi Almaghrabi, the director of planning and quality, and Adnane Habib, the quality head of the sector. They have made a considerable effort to establish the education sector's quality system and spread awareness among all staff and students [30,31]. We learned from their effort and knowledge; hence, we were able to write this paper in the hopes of contributing to the state of the art in the quality control of academic establishments.

**Conflicts of Interest:** The authors declare no conflict of interest.

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
