# Peer review of "Use Dynamic Scheduling Algorithm to Assure the Quality of Educational Programs and Secure the Integrity of Reports in a Quality Management System"

_information, doi:10.3390/info12080315_

Round 1

Reviewer 1 Report

The major issues of this paper are that,

  1. “Figure 3.” Should be changed into “Table ..”.
  2. In your paper, the table name and number should be put at the top of the table.
  3. Please highlight your contributions or innovations.

Author Response

Answer to the first review:

The major issues of this paper are that,

  1. “Figure 3.” Should be changed into “Table ..”.

The change was made as shown in page (7 )

  1. In your paper, the table name and number should be put at the top of the table.

We did the change on all the tables of the manuscript

  1. Please highlight your contributions or innovations.

We updated the abstract and improved the introduction to highlight the contribution check pages (1-2)

Additional changes :

English language and style:

We review the paper and use electronic editors to detect issues and improved the style of writing in multiple places of the paper.

Are the methods adequately described?

We improved the methodology part (page 5-7)

We improved the introduction part and made significant change to highlight the motivation (page1-2).

Is the research design appropriate?

We improved the methodology part (page 5-7)

Are the results clearly presented?

We added a discussion section to provide more explanation on the findings. (page 10 )

Are the conclusions supported by the results?

We provided discussion section to explain the results. (page 10 )

Reviewer 2 Report

Main question addressed by the research is how to apply the Tomasulo algorithm to control the processes of academic program operations. 

The topic of this paper is very actual and interesting, there is strong pressure in the world to increase the quality of educational institutions and the output of this article in the form of a case study, can be useful for educational institution around the world. 

Tomasulo algorithm was used to allow a single processor to operate sequential code instructions and exploit the parallelism. The authors not only explain how the algorithm is used for the microprocessor, but also identified similarities between the operations of the microprocessor environment and the operation in workplaces. Based on the identification of these similarities, authors explained the use of the algorithm in an academic environment. 

The paper is easy to read.  

Author Response

(The authors gave the same response as above.)

Round 2

Reviewer 1 Report

The authors have enhanced the manuscript by far. Minor issues are that many heuristic or metaheuristic methods for the problems with uncertain factors, such as evolutionary multi-objective blocking lot-streaming flow shop scheduling with machine breakdowns, multi-objective migrating birds optimization algorithm for stochastic lot-streaming flow shop scheduling with blocking, should be stated in Section 1. Except for the above issues, I have no more comments.

Author Response

Dear sir 

I am not quite sure if I understand your comments in this review, 

I will improve the English and make more corrections for the next round. 

Is there any other comments? 

thanks